# Transformative optimisation of agricultural land use to meet future food demands

Lian Pin Koh[1], Thomas Koellner[2] and Jaboury Ghazoul[1]

[1] Department of Environmental Systems Science, ETH Zurich, Zurich, Switzerland
[2] Faculty of Biology, Chemistry and Geosciences, University of Bayreuth, Bayreuth, Germany

## ABSTRACT

The human population is expected to reach ∼9 billion by 2050. The ensuing demands for water, food and energy would intensify land-use conflicts and exacerbate environmental impacts. Therefore we urgently need to reconcile our growing consumptive needs with environmental protection. Here, we explore the potential of a land-use optimisation strategy to increase global agricultural production on two major groups of crops: cereals and oilseeds. We implemented a spatially-explicit computer simulation model across 173 countries based on the following algorithm: on any cropland, always produce the most productive crop given all other crops currently being produced locally and the site-specific biophysical, economic and technological constraints to production. Globally, this strategy resulted in net increases in annual production of cereal and oilseed crops from 1.9 billion to 2.9 billion tons (46%), and from 427 million to 481 million tons (13%), respectively, without any change in total land area harvested for cereals or oilseeds. This thought experiment demonstrates that, in theory, more optimal use of existing farmlands could help meet future crop demands. In practice there might be cultural, social and institutional barriers that limit the full realisation of this theoretical potential. Nevertheless, these constraints have to be weighed against the consequences of not producing enough food, particularly in regions already facing food shortages.

## INTRODUCTION

By 2050, the global human population will have grown from the current ∼7 billion to ∼9 billion people (*United Nations, 2008*). These people will require more food (*Evans, 2009*; *Godfray et al., 2010*). They are also likely to demand a higher proportion of meat and dairy products that require more land, water and energy to produce (*Royal Society of London, 2009*; *Tilman et al., 2001*). Meeting this demand is daunting by virtue of the need to reduce greenhouse-gas emissions (*Meinshausen et al., 2009*), minimise fertiliser and pesticide inputs (*Moss, 2007*), and avoid further impacts on natural ecosystems and wildlife (*Ehrlich & Pringle, 2008*). Additionally, we might have to cope with the yet unclear implications of climate change on food security (*Brown & Funk, 2008*; *Lobell et al., 2008*; *Parry et al., 2004*).

Corresponding author
Lian Pin Koh,
lianpinkoh@gmail.com

These challenges might be met by closing yield gaps (i.e., difference between potential and actual yields) or raising yield ceilings, reducing food lost to waste, and switching to less protein-rich or more aquaculture-based diets (*Foley et al., 2011*; *Godfray et al., 2010*).

Additionally, we propose that a complementary approach is to maximise agricultural returns by planting crops that are best suited to site-specific conditions. While this strategy might seem obvious, the degree to which agricultural land use is optimised and the benefits of optimisation have not been evaluated at a global scale by which benefits might be maximally realised. To test the efficacy of this land-use optimisation approach, we developed a spatially-explicit computer simulation model based on the following algorithm: on any cropland, always produce the most productive crop given all other crops currently being produced locally and the site-specific biophysical, economic and technological constraints to production. By evaluating crops based on their realised yields, the algorithm captures both the local biophysical limitations to production (e.g., the need for irrigation), and the behaviour of farmers in response to these constraints (e.g., the decision to irrigate or not). Therefore, for a farmer who is currently growing barley, maize, wheat and irrigated rice on his land, and if irrigated rice has the highest per-hectare realised yield given local conditions, then land-use optimisation would entail devoting the entire farmland to irrigated rice production. An implicit requirement of this approach is that goods being considered are fungible, such that individual units of different crops within a commodity group (e.g., cereals or vegetable oil) are mutually substitutable. Therefore, we illustrate our approach by optimising land use within each of two groups of essential and fungible food crops: cereals (barley, maize, millet, rice, sorghum and wheat) and oilseeds (soy, cottonseed, rapeseed, sunflower seed, groundnut and oil palm). We optimised land use by replacing all currently harvested area, for cereals or oilseeds, with the most productive crop in the set of currently harvested crops within each farmland (Fig. 1) (*Monfreda, Ramankutty & Foley, 2008*).

## MATERIAL AND METHODS

We assessed geospatial information on current land-use and crop-yield for these crops at the farmland scale across 173 countries (*Monfreda, Ramankutty & Foley, 2008*). We based our analyses on a published global geospatial dataset at 5 arc-minute resolution ($\sim$10 $\times$ 10 km grid cell) that depicts, for the year 2000, the proportion of harvested area and actual yield reported for each crop in each grid cell (*Monfreda, Ramankutty & Foley, 2008*). We overlaid these data to produce a new data layer of intersected polygons (i.e., land areas sharing unique geospatial information on observed yield for each crop; referred to as farmlands in text). For cereals, these data encompass a total area of 651 million ha ($\sim$42% of Earth's total arable and permanent croplands) (*FAO, 2012*) and comprise 788,557 data polygons (polygon mean area $= 826 \pm 4.1$ ha [$\pm$standard error]); for oilseeds, these data encompass a total area of 184 million ha and comprise 426,000 data polygons (mean area $= 433 \pm 2.9$ ha).

We carried out a three-step procedure to estimate optimised crop production within each farmland (Fig. 1). First, we established a baseline of current total production of

**Peer**J

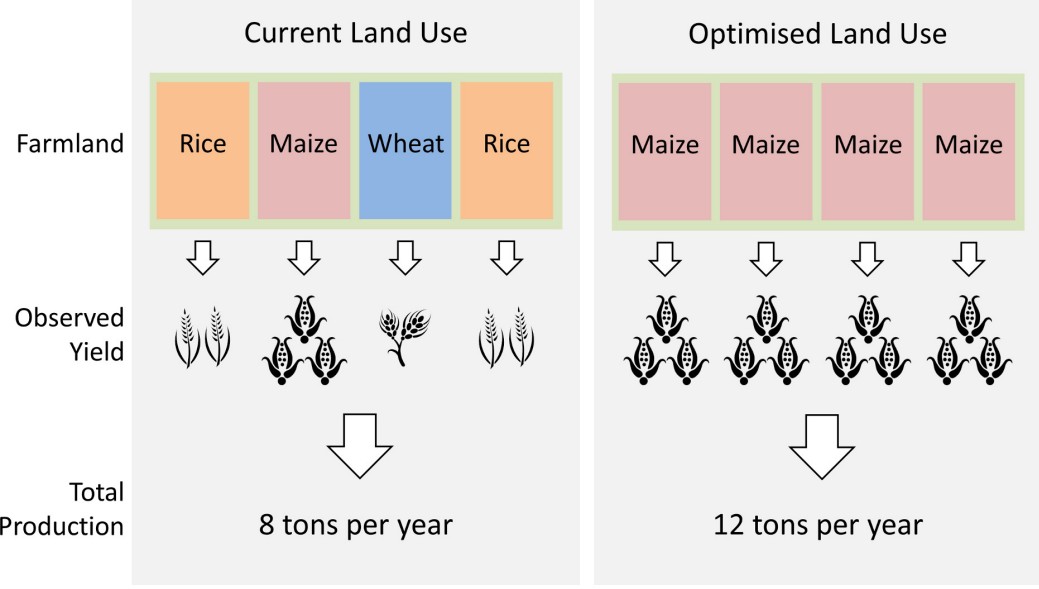

**Figure 1 Land use optimisation.** We optimised land use by replacing all currently harvested area, for cereals or oilseeds, with the most productive crop in the set of currently harvested crops within each farmland. This cartoon illustrates optimisation of cereal crops within a farmland: by converting the entire farmland for production of the optimal crop (maize in this case), total cereal production increased by 4 tons per year.

cereal or oilseed crops within each farmland. We did so by multiplying harvested area with observed yield of each cereal or oilseed crop (*Monfreda, Ramankutty & Foley, 2008*). Unlike cereals, whereby cereal grain is the prime economically-important product, oilseeds are produced for both oil and meal. We calculated vegetable oil and protein meal production amounts by multiplying crop production with an oil- or meal-conversion factor (derived from 2008/09 data on global crop, oil and meal production) (*USDA-FAS, 2011*).

Second, we identified an optimal cereal or oilseed crop within each farmland. The optimal cereal crop was the one with the highest observed yield within each farmland. In identifying an optimal oilseed crop, we assessed relative productivity based on the combined quantity of oil and meal produced. Given that global demand for protein meal is higher than that for vegetable oil, in optimising for oilseed production, we ascribed meal a relative weightage of 1.77 tons for every ton of oil produced (derived from 2008/09 data on global crop, oil and meal consumption) (*USDA-FAS, 2011*).

Third, we simulated land use optimisation by converting each farmland to a mono-culture of the identified optimal cereal or oilseed crop. We calculated the resultant cereal or oilseed production for each optimised farmland by multiplying total harvested area with observed yield of the optimal crop. We calculated the benefits of land use optimisation by comparing this new production volume with the baseline production prior to optimisation.

In our analyses we made the following assumptions. First, we assumed that the reported yield of each crop is uniform within each farmland (mean area = 826 ± 4.1 ha for cereals and 433 ± 2.9 ha for oilseeds). If parts of a farmland had substantially lower

PeerJ ___________

(or higher) than the reported yield for the optimal crop, we would have overestimated (or underestimated) the benefits of land use optimisation. Second, we assumed that site-specific biophysical, economic and technological constraints to production are also uniform at the scale of each farmland, such that optimisation of a farmland for any optimal crop would not be limited by, for example, variations in water scarcity or soil nutrient levels across different parts of a farmland.

## RESULTS

Globally, our strategy resulted in net increases in annual production of cereal and oilseed crops from 1.9 billion to 2.9 billion tons (46%), and from 427 million to 481 million tons (13%), respectively, without any change in total land area harvested for cereals (651 million ha) or oilseeds (184 million ha) (Tables S1–S4). Accordingly, annual production of vegetable oil and protein meal (the primary products of oilseeds) increased from 86 million to 94 million tons (10%), and from 176 million to 228 million tons (29%), respectively. Global demand for cereals is projected to increase to 2.7 billion tons by 2030, and to 3 billion tons by 2050 (including its use as animal feed) (*FAO, 2006*). As such, land-use optimisation could contribute substantially to meeting future demands for cereals (at least until 2030). In contrast, the modest benefits of optimisation for vegetable oil production would not be sufficient to meet expected demands in 2030 (216 million tons) or 2050 (293 million tons) (*FAO, 2006*).

Among cereal crops, maize and rice underwent the largest expansions in the harvested area, accompanied by increases in annual crop production by 746 million tons and 560 million tons, respectively (Fig. 2A). All other cereal crops, with the exception of sorghum, declined in both area (by at least 50%) and production (by 44%–53%; Fig. 1A). Although the harvested area for sorghum declined by 16 million ha (40%), annual production increased by 22 million tons (38%). This is due to an increase in sorghum's average annual yield from 1.7 to 3.4 tons/ha, as a result of land-use optimisation (Fig. S1).

In the case of oilseeds, soy was the only crop that expanded in area (by 60 million ha or 81%; Fig. 2B). Soy was also the only oilseed crop to experience a decrease in average annual yield (from 2.4 to 2.2 tons/ha), as most of its expansion occurred on lands that were sub-optimal for soy but still more productive under soy than under any other crop (Fig. S2). Even so, annual production of soy increased by 98 million tons (60%). Oil palm production also increased by 23 million tons (20%; Fig. 2B).

We next assessed whether the benefits of land-use optimisation would be manifested where most required, by exploring its implications for cereal production in five regions of the world that face the most severe food shortages, and would likely continue to do so in the future. These regions, which include South Asia, China, Southeast Asia, East Africa and Central Africa, contain 75% (657 million) of the world's malnourished people (*FAO, 2012*; *Lobell et al., 2008*). We found that in South Asia, China and Southeast Asia, rice would remain the dominant cereal crop (Fig. 3). In fact, rice-cultivated areas would increase from 129 million to 176 million ha at the expense of wheat, millet and sorghum which, incidentally, are thought to be the most vulnerable to climate change impacts in South Asia

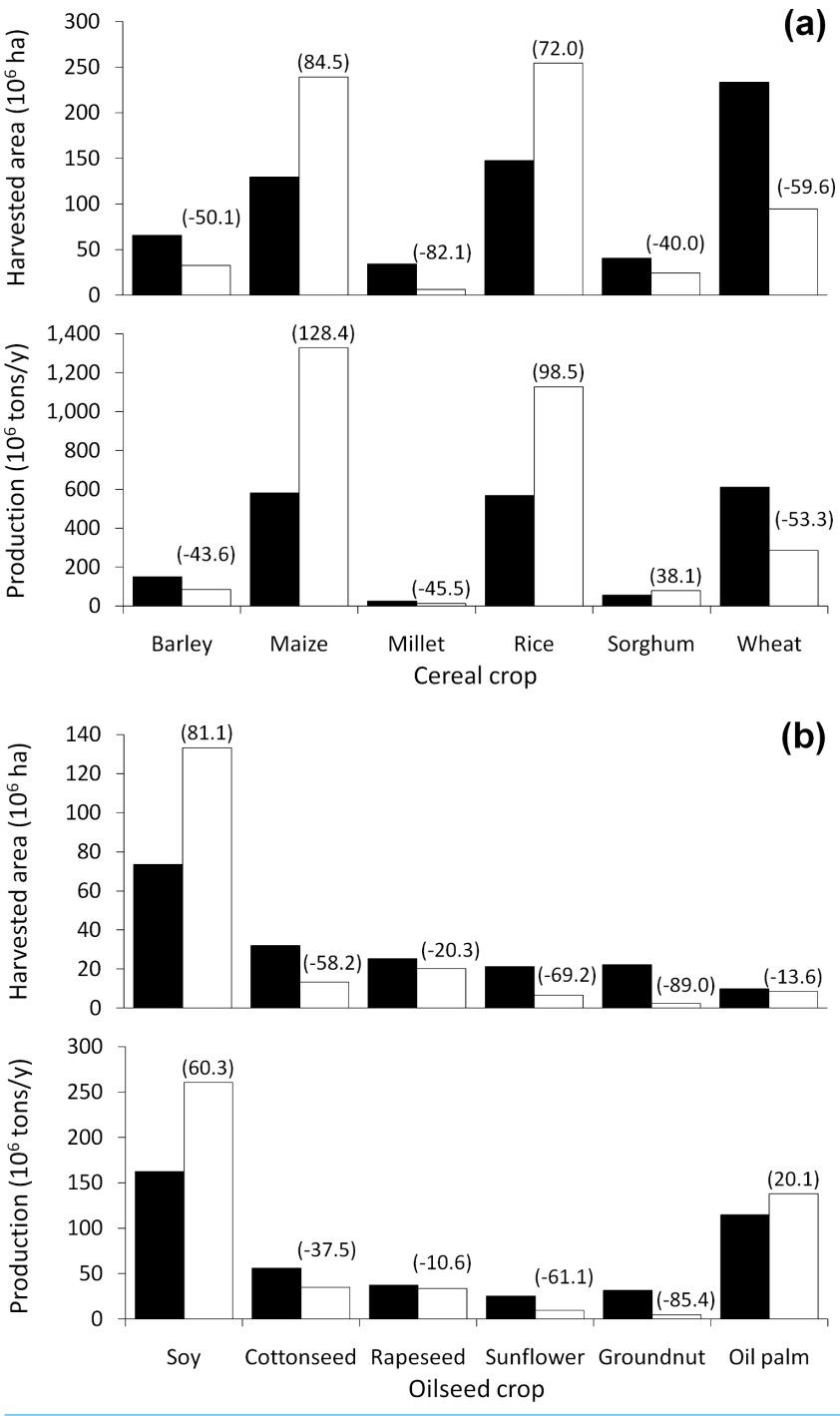

**Figure 2  Harvested area and crop production.** Changes in cultivated area and production amounts of (A) cereal (171 countries) and (B) oilseed crops (167 countries) under current and optimal land-use allocations. Filled bars represent current land use; open bars represent optimal land use; and numbers in parentheses indicate percentage change.

**Peer**J

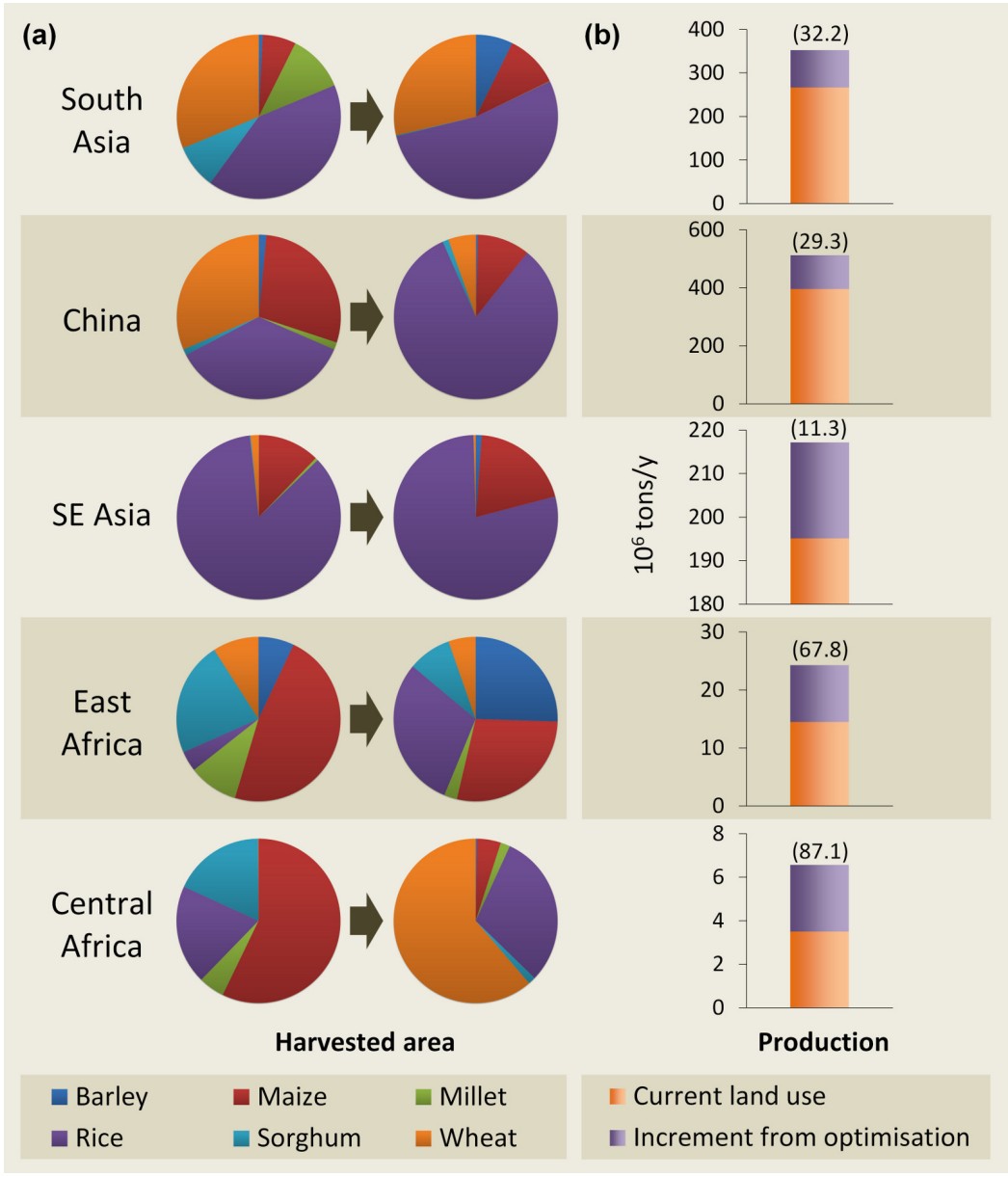

**Figure 3 Benefits of land-use optimisation in food-insecure regions.** (A) changes in relative proportions of harvested area of cereal crops; (B) changes in total crop production (numbers in parentheses indicate percentage change).

(*Lobell et al., 2008*) (Fig. 3). In East Africa and Central Africa, maize would no longer be the dominant cereal crop. Instead, East Africa would grow mainly rice (3.1 million ha), maize (3 million ha) and barley (2.7 million ha), while Central Africa would specialise in wheat (2.2 million ha) and rice (1.1 million ha) (Fig. 3). Following land-use optimisation, Central Africa would almost double its annual production of cereals (Fig. 3). The other four regions would also experience increases in cereal production (11%–68%) (Fig. 3).

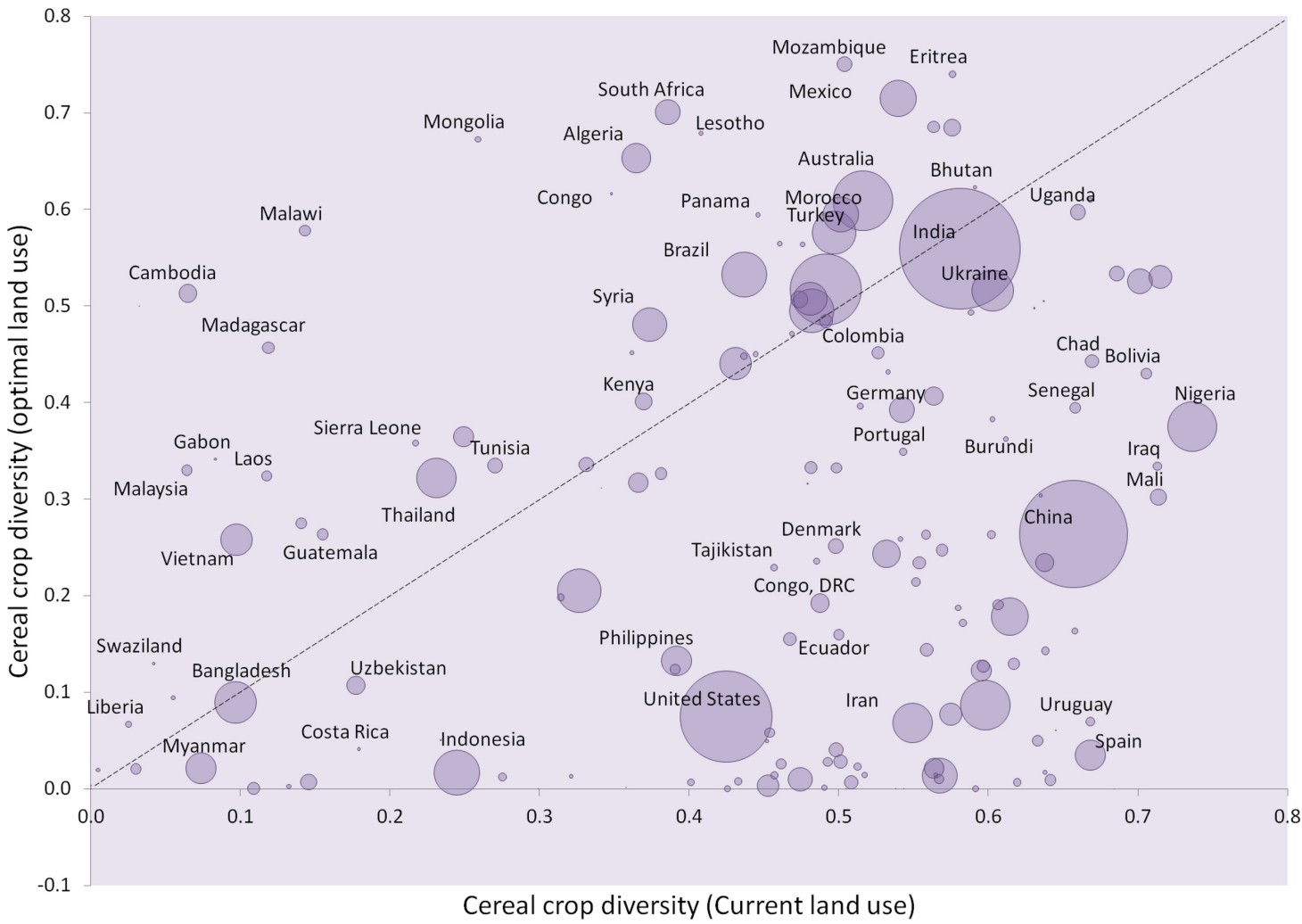

**Figure 4 Cereal crop diversity.** Crop diversity was calculated based on the reciprocal Simpson's diversity index, $1 - \sum p_i^2$, whereby $p$ represents the proportional area of the $i$th crop type. This index reflects the probability that two randomly chosen cropland areas are not cultivated for the same cereal crop. Circle areas reflect relative total harvested area for cereal crops. Dashed line indicates no change in crop diversity between current and optimal land uses. For clarity of presentation, not all country labels are shown.

## DISCUSSION

We recognise that besides productivity, other cultural and socio-political considerations also determine actual land use and production systems. For example, land-use optimisation entails reducing annual rice production in Thailand from 24.4 million to 5.9 million tons (Table S3). Rice farmers in Thailand might be hesitant to switch to planting other cereal crops as rice has a long history of cultivation and consumption in the region, in the same way that maize is intimately associated with the cultures and history of the Americas. To explore the effects of such cultural constraints, we re-ran the model with a modified algorithm, which excluded rice-cultivated areas from the optimisation process. In this case optimised global annual production for cereals was 2.7 billion tons, slightly less than

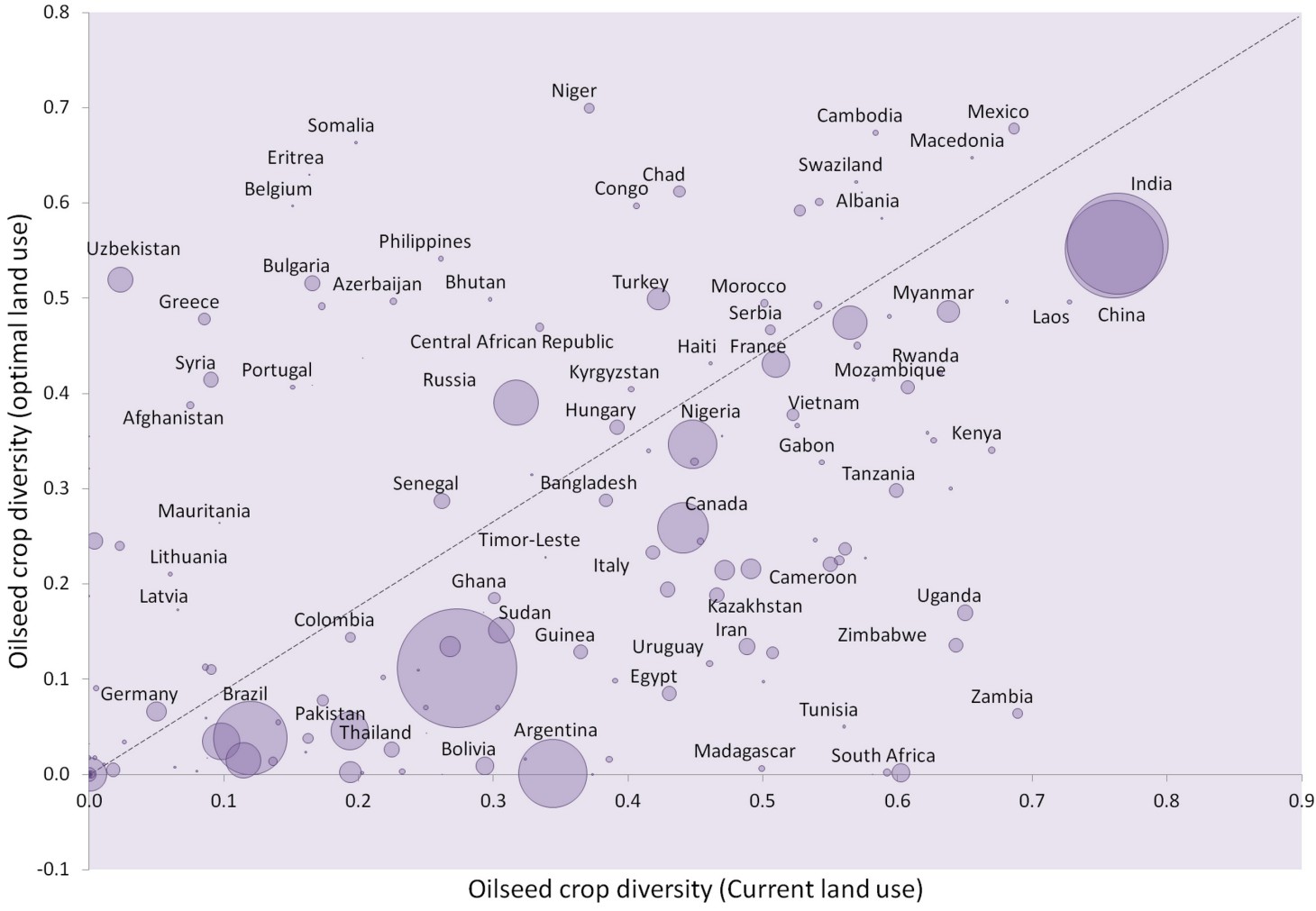

**Figure 5  Oilseed crop diversity.** Crop diversity was calculated based on the reciprocal Simpson's diversity index, $1 - \sum p_i^2$, whereby $p$ represents the proportional area of the $i$th crop type. This index reflects the probability that two randomly chosen cropland areas are not cultivated for the same oilseed crop. Circle areas reflect relative total harvested area for oilseed crops. Dashed line indicates no change in crop diversity between current and optimal land uses. For clarity of presentation, not all country labels are shown.

the 2.9 billion tons projected based on the optimisation of all six cereal crops. Thus the exclusion of rice from optimisation has little overall impact on the production of cereals.

The specialisation of production systems implies homogenisation of farms and agricultural landscapes. Yet some farmers might prefer to maintain multiple crops for various reasons, including balancing dietary requirements, and bet-hedging against outbreaks of pests and diseases, adverse weather conditions and price fluctuations that a monoculture might be more sensitive to. While land use optimisation might indeed drive homogenisation within individual farmlands, it is not necessarily so at national and regional scales: there is considerable variation in crop diversity following optimisation, with diversity actually increasing in many countries and regions (Figs. 3–5). We do not necessarily advocate that nations should pursue, solely, a production maximization strategy, but rather our results indicate the potential for substantial increases in crop

production with such an approach. In practice, other considerations, including the benefits of maintaining diverse cropping systems, will necessarily affect the agricultural decisions taken. Neither do we imply that land-use optimisation is the only solution. On the contrary, a move towards optimisation should be implemented alongside other solutions, such as closing yield gaps, which are especially high for maize in Sub-Saharan Africa (*World Bank, 2008*). In fact, land-use optimisation needs to be complemented by improvements in farming technologies and institutional structures, such as education, and market and financial risk management systems, all of which farmers need to make best use of the land and technologies available to them. Furthermore, given that smallholder farming often is the most common form of agricultural organisation, especially (but not only) in the tropics, smallholders will need to be integrated in any land-use optimisation approach through the provision of education, technology, and market and finance opportunities.

In conclusion, our assessment demonstrates that in theory future crop demands, at least for cereals, can be substantially met on existing agricultural land area through the pursuit of more optimal use of farmlands. In practice there might be cultural, social and institutional barriers that limit the full realisation of this theoretical potential. Nevertheless, these constraints have to be weighed against the consequences of not producing enough food, particularly in regions already facing food shortages.

### Funding
Lian Pin Koh is supported by the Swiss National Science Foundation. The funder had no role in study design, data collection and analysis, decision to publish, or preparation of the manuscript.

### Grant Disclosures
The following grant information was disclosed by the authors:
Swiss National Science Foundation.

### Competing Interests
The authors state that there are no competing interests.

### Author Contributions
- Lian Pin Koh conceived and designed the experiments, performed the experiments, analyzed the data, contributed reagents/materials/analysis tools, wrote the paper.
- Thomas Koellner and Jaboury Ghazoul contributed reagents/materials/analysis tools, wrote the paper.

### Supplemental Information
Supplemental information for this article can be found online at http://dx.doi.org/ 10.7717/peerj.188.

**Peer**J

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
