# Peer review of "Transformative optimisation of agricultural land use to meet future food demands"

_PeerJ, doi:10.7717/peerj.188_

## Round 0.1 · original submission · Minor Revisions

Please respond to the reviewers comments by amending the manuscript. Respond positively where you are able and include discussion of the issue where you disagree with views expressed by the reviewers. I draw particular attention to the issue of scale identified by reviewer two and exchangeability (fungability?) by reviewer three. Please ensure that there is appropriate discussion of the potential impacts of both on your conclusions.

·

Basic reporting

No comment

Experimental design

Please could the authors clarify the following?

How is the optimal yield for each crop/land area calculated & what assumptions are made/what factors are taken into account?

What countries/regions are included in Figure 1? Is it all regions (including western Europe, Canada, US etc. as Figures 3&4 appear to show, or only the 'food insecure' regions specified in Figure 2?

Also, the data set used for 'current' land use appears to be 13 years old. Are the authors aware of how land-use/farming systems may have changed in any regions during that period & of any impacts said changes may have on their results/conclusions?

Validity of the findings

This is an interesting concept, the implications of which are perhaps oversimplified in the conclusions.

Although the authors recognise that there may be limitations to optimisation, I am concerned that as it stands at the moment some of the most important practical limitations (and potential impacts) are not acknowledged, and need to be discussed.
These include limitations caused by water scarcity (and any potential impacts of the proposed model on water security), any requirement for rotations to manage productivity, trade limitations/impacts, environmental impacts & any impact on nutritional deficiencies.

I am unsure whether the statement in the final sentence is directly relevant to the findings and would be more comfortable with conclusions that encouraged elements of the optimisation approach to be adopted where feasible.

·

Basic reporting

I must stress that I am not an expert on agricultural ecosystems or crop yields and what constrains them. That said, I'm interested in global environmental issues and understand both estimates of production and diversity that form the core of this paper. I accepted the chance to review this manuscript because I have followed the senior author's work closely and admire his creative approaches to many problems.

This manuscript addresses a disarmingly simple question: can we grow more crops if we adjusted the mix to maximise productivity. The answers answer an emphatic "yes" for cereals while improvements for oilseeds are much smaller. All that said, the most interesting aspects are why countries have not optimised productivity. The authors suggest a variety of possibilities. At the risk of asking them to expand a paper that is short and to the point, it seems that a more complete examination of what limits optimal production is warranted.

1. I consider the issue of spatial scale in the next section.

2. Would an optimal production lead to greater profits for the farmers? And to what extent are allocations driven by national subsidies?

3. Crop diversity is important. I found figures 3 and 4 to be most informative. Clearly most production is either close to optimal diversity or exceeds it considerably. The USA and China, for example, would need to move towards much less diverse croplands if they were to improve cereal production. Spain would need to become a cereal crop monoculture, for example.

4. Large food producers are unlikely to wish to modify current allocations to feed other countries. I would like to see the improvements of production ranked by the net balance of food exports and imports. Could food importers avoid such dependency? And at what cost in terms of crop specialization?

Experimental design

1. My first worry is about scale. 10 x 10km is fine scale, certainly, but I'd like to be reassured that the following possibility is excluded. For such a pixel, quite possibly irrigated rice may attain the highest productivity within a small piece of that. Extrapolating such productivity across the pixel may be impossible. Just think of what happens along the Nile, for example, where one can stand with one foot in very productive crops and the other in desert. Yes, irrigated rice is more productive than rainfed wheat, but that doesn't mean one can grow rice everywhere within the pixel.

2. The largest changes proposed would be to replace maize with wheat in Central Africa — for a huge increase in production — and to reduce wheat in China, but grow more rice. The authors mention these changes (page 5), but do not further investigate why the changes haven't been made. Water may well prevent rice from replacing wheat in China, and soil nutrients (and water) may well prevent wheat from replacing maize in Africa, especially one considers the scale issues I have already mentioned.

The way to investigate these possibilities is to examine a sample of pixels that seem particularly suboptimal —where, for example, rice production is high per unit area within the pixel, but only a small fraction of the pixel grows rice. If that's an irrigation issue, then the authors need to assess how large an error this causes.

Validity of the findings

See concerns expressed above.

Additional comments

I view this as being most interesting as a way of documenting what the limitations are to increased production. The bottom line — substantial improvements — are subject to many caveats. The value of this manuscript is to list what some of them are.

·

Basic reporting

The article is clearly written, interesting, and well prepared. The figures support the reported conclusions.

A few points of clarification are needed:

1) Introduction, line 1: According to the UN Population Division, the human population exceeded 7 billion in October 2012 (on Halloween, notably, although this is obviously just an approximation).

2) Introduction, on the assumption of the fungibility of crops: Obviously, this is quite a large assumption in the context of the present analysis. One could imagine lots of reasons for farmers electing to have multiple crops, ranging from balancing their dietary requirements to bet-hedging against crop-specific pathogens, weather, and crop-price fluctuations. Some brief discussion of this later in the paper would be warranted.

3) Results section: One point on which I was not clear was crop transport. Some crops might be produced near to where their demand is concentrated, even if that locale is suboptimal. Is this factored into the analysis? I presume not. Again, this might be mentioned briefly in the Discussion.

Experimental design

The design of the analysis is effective and well considered, and falls within the scope of the journal. The paper contains a great deal of interesting analysis and interpretation.

Validity of the findings

My sense is that the analyses are reasonably robust and effectively interpreted, using the best available information and data sets at hand. The conclusions seem broadly justified by the analyses, notwithstanding the need for some minor points of clarification as indicated above.

Additional comments

I found much of interest in this paper. It is appropriately framed as a sort of thought experiment, and addresses some very big and important questions.

---

## Round 0.2 · accepted · Accept

Your response to the reviewers questions have addressed substantive concerns and improved the manuscript. Thank you for engaging positively with the peer review process.